# Low Serological Agreement of Hepatitis E in Immunocompromised Cancer Patients: A Comparative Study of Three Anti-HEV Assays

**DOI:** 10.3390/antib14020027

**Published:** 2025-03-24

**Authors:** Isabel-Elena Haller, Mark Reinwald, Janine Kah, Franz A. M. Eggert, Sandra Schwarzlose-Schwarck, Kristoph Jahnke, Stefan Lüth, Werner Dammermann

**Affiliations:** 1Department of Gastroenterology, University Hospital Brandenburg, Brandenburg Medical School Theodor Fontane, 14770 Brandenburg an der Havel, Germany; 2Center of Translational Medicine, Brandenburg Medical School Theodor Fontane, 14770 Brandenburg an der Havel, Germany; 3Faculty of Health Sciences, Joint Faculty of the Brandenburg University of Technology, Brandenburg Medical School and University of Potsdam,14469 Potsdam, Germany; 4Department of Hematology and Oncology, University Hospital Brandenburg, Brandenburg Medical School Theodor Fontane, 14770 Brandenburg an der Havel, Germany; 5Department of Internal Medicine, University Medical Center Hamburg-Eppendorf, 20246 Hamburg, Germany; 6Department of Neurosurgery, School for Mental Health and Neuroscience, Maastricht University, 6229 ER Maastricht, The Netherlands; 7Oncology Specialist Practice Brandenburg, 14772 Brandenburg an der Havel, Germany

**Keywords:** anti-HEV antibodies, immunosuppression, oncology, enzyme-linked immunosorbent assay (ELISA), electrochemiluminescence immunoassay (ECLIA)

## Abstract

Background/Objectives: Hepatitis E virus (HEV) is one of the leading causes of acute hepatitis, with immunosuppressed individuals, such as oncology patients, being particularly vulnerable to chronic infections that may progress to liver disease or fatal outcomes. Assay variability complicates HEV prevalence assessment in at-risk groups. This study aimed to compare the reliability and concordance of three HEV antibody assays—Wantai, Euroimmun, and Elecsys^®^—in immunosuppressed oncology patients. Methods: In this prospective pilot study, serum samples were obtained from oncology patients between September 2020 and October 2021. Samples were collected both at baseline (treatment-naive) and during ongoing treatment. A healthy control group was retrospectively included for comparative analysis. Anti-HEV IgM and IgG antibodies were tested in all samples using enzyme-linked immunosorbent assays (Wantai, Euroimmun) and an electrochemiluminescence immunoassay (Elecsys^®^). Demographic and clinical data, along with information on HEV risk factors, were extracted from medical records and patient questionnaires. Results: HEV IgM prevalence ranged from 0% (Wantai) to 6% (Elecsys^®^), while IgG prevalence was 12% (Euroimmun), 38% (Wantai), and 53% (Elecsys^®^). Concordance was poor, with Cohen’s Kappa values indicating slight to moderate agreement (κ = 0.000–0.553). Patients with hematological malignancies exhibited the highest IgG seroprevalence. Risk factor analysis revealed the highest association between HEV exposure and the consumption of undercooked pork or crop-based agriculture. Conclusions: Significant variability among HEV serological assays highlights the challenges of reliable HEV diagnostics in immunosuppressed oncology patients. Assay selection and improved testing strategies are critical for this high-risk group.

## 1. Introduction

Hepatitis E virus (HEV) represents one of the most common causes of acute hepatitis [1]. The spectrum of clinical manifestation varies and is particularly influenced by patients’ immune status and the pathogens’ genotype (GT) [2]. HEV etiology varies geographically: While in Asia, Africa and parts of Latin America fecal–oral transmission is most common, the anthropozoonotic HEV-strain GT3, whose main reservoirs are pigs, wild boars and deer, is the most prevalent in Germany [3,4]. In this context, close contact with infected animals and consumption of contaminated food, especially raw or undercooked meat, have been identified as major routes of transmission [3,5,6]. Indirect transmission, including the use of contaminated water for fruit and vegetable washing or fertilizer made from animal feces, has also been reported [7,8,9].

Compared to immunocompromised patients, healthy individuals who acquire zoonotic HEV infections often remain asymptomatic and experience a self-limiting disease course [10]. Against this background, immunosuppression characterizes a known risk factor for chronic HEV infection [11,12]. At-risk groups include patients with a history of solid organ transplantation (SOT) [13], patients with solid tumors, patients with hematologic malignancies receiving antineoplastic medication such as chemotherapy [14], people living with human immunodeficiency virus (HIV) [15] and those with autoimmune diseases receiving immunomodulatory drugs [16]. Hypothesized and researched mechanisms underlying this increased risk amongst these groups include dysregulated interferon response [17], lower T cell counts [13] and mobilization of gamma delta T cells [18], which contribute to higher morbidity, characterized by higher progression rates of acute infections into chronic liver disease [5] as well as higher morbidity. Indeed, one retrospective multicenter cohort study of hematological patients across 11 European centers reports a positive association between persistent RNA-positivity for HEV and an excess mortality of 16% [19].

Chronic hepatitis E is defined as the persistence of detectable HEV-RNA in serum or stool for more than 3 months and was first reported in SOT recipients in France [13]. Since then, it has also been reported in other patients receiving immunocompromising therapies such as corticosteroids, cyclosporine, methotrexate and rituximab [20,21,22,23,24,25,26].

Amongst these individuals, including SOT recipients, patients with hematological malignancies and those who have undergone bone marrow transplantation, increased risk of developing liver cirrhosis and liver failure requiring transplantation have been reported [13,27].

The detection of anti-HEV antibodies in serum plays a critical role in the diagnosis of HEV. Numerous studies have highlighted significant discrepancies in HEV seroprevalences depending on the specific anti-HEV assay used [28,29,30]. Even within the same study population, substantial variability in seroprevalence has been observed [10,31,32]. As a result, the performance of widely used anti-HEV assays has been evaluated across various cohorts in several studies. However, it is important to note that the majority of these studies primarily focused on immunocompetent patients. There is a notable lack of studies assessing the performance of different anti-HEV assays in immunosuppressed oncology populations, leaving this at-risk patient cohort underrepresented. Therefore, the aim of this prospective pilot study was to evaluate the reliability of three anti-HEV assays (Wantai, Euroimmun, Elecsys^®^) in a cohort of immunosuppressed oncology patients. While the Wantai and Euroimmun assays are well-established ELISA tests frequently cited in the literature, no data is currently available on the Elecsys^®^ HEV immunoassay in an immunocompromised population.

## 2. Methods

### 2.1. Patients

The study was prospectively designed as a pilot trial and conducted at Brandenburg University Hospital in the German federal rural state of Brandenburg between September 2020 and October 2021. It was part of a broader HEV research project spanning three federal states in Germany: Brandenburg, Schleswig-Holstein, and Hamburg. Ethical approval was obtained from the ethics committees of the medical boards in all three states (reference number E-01-20200522).

Adult patients with various hematooncological conditions (including leukemia, lymphoma, and solid tumors) were included without age restrictions, provided they gave written informed consent. Patients who were already undergoing antineoplastic therapies were excluded. Demographic and clinical information was retrieved from the hospital’s internal clinical information system. Serum samples were collected prior to the initiation of immunosuppressive therapy and during follow-up visits while patients received ongoing treatment.

Surplus serum samples from healthy controls, including healthcare workers from the gastroenterology and cardiology departments at Brandenburg University Hospital, University Medical Center Hamburg-Eppendorf, and Municipal Hospital Kiel, were analyzed for anti-HEV-IgM and IgG seroprevalence and acted as controls. However, due to the limited availability of samples, age- and sex-matching between the oncological cohort and the healthy controls was not feasible.

### 2.2. Anti-HEV Antibody Measurement

HEV antibodies were qualitatively measured in serum samples using three analytical methods: (a) the Wantai enzyme-linked immunosorbent assay (ELISA), considered the gold standard, which included the WANTAI HEV-IgM and IgG kits (Beijing Wantai Biological Pharmacy, China); (b) the Euroimmun Anti-HEV IgM and IgG assays (EUROIMMUN Medizinische Labordiagnostika AG, Germany); and (c) the Elecsys® Anti-HEV IgM and IgG kits (Roche Diagnostics GmbH, Basel, Switzerland). All assays were performed according to the manufacturers’ instructions. The analysis and serostatus interpretation were based on the cut-off values provided by the manufacturer.

For the Wantai- and the Euroimmun-ELISA, results were expressed as the ratio between the sample’s optical density (OD) and the cut-off OD. For the Wantai assays, IgM and IgG ratios below the cut-off value of 1, including borderline values between 0.9 and 1.1, were classified as negative, while ratios exceeding 1 were classified as positive. For the Euroimmun assays, ratios between 0.8 and 1.1 were considered borderline, ratios below 1.1 as negative, and those equal to or above 1.1 as positive.

To address serological discrepancies between the results from the Wantai and Euroimmun assays, we introduced a third testing approach. The Elecys^®^-assay is a fully automated chemiluminescence immunoassay, performed on the cobas analyzer platform, which employs recombinant antigens to detect HEV IgM and IgG antibodies in vitro. For IgM testing, a cut-off index (COI) below 1 indicated non-reactivity, while a COI equal to or above 1 indicated reactivity. For IgG testing, non-reactivity was defined as values below 0.15 units per milliliter (U/mL), and reactivity was defined as values equal to or above 0.15 U/mL.

### 2.3. Questionnaire

A seven-item, non-validated questionnaire was administered to oncological participants. The items were designed based on established risk factors for HEV to evaluate their potential relevance in immunocompromised patients. The questions were formatted using dichotomous or Likert scale responses (Appendix A).

### 2.4. Statistical Analysis

Categorical variables were analyzed using the non-parametric Chi-Square test and Fisher’s Exact test. Cohen’s Kappa (κ) statistic was employed to assess the level of agreement between the two assays. For the level of agreement, we followed the criteria for the interpretation of kappa values proposed by Landis and Koch (1977) [33]. According to these criteria, a Cohen’s kappa coefficient between 0.00 and 0.20 indicates slight agreement, between 0.21 and 0.40 fair agreement, between 0.41 and 0.60 moderate agreement, between 0.61 and 0.80 substantial agreement, and values greater than 0.80 represent almost perfect agreement.

Statistical significance was defined as *p* < 0.05. In addition to descriptive statistics, univariate logistic regression was performed to analyze the questionnaire data and data provided by the Wantai assay, with odds ratios (OR) and 95% confidence intervals (CI) reported for each variable. All statistical analyses were conducted using IBM SPSS Statistics (version 29.0.0.0, Chicago, IL, USA) and GraphPad Prism (version 10.1.0, Boston, MA, USA).

## 3. Results

### 3.1. Study Population

A total of 66 immunocompromised individuals (oncological patients) and 65 healthcare workers (healthy controls) were included in the analysis. Demographic characteristics of both cohorts, along with clinical details of patients with oncological diseases, are summarized in Table 1. For patients undergoing oncological treatment, serum samples were collected at baseline and during follow-up visits. Several factors such as mortality, treatment discontinuation, or transfer to another hospital led to patient dropouts during follow-up. Specifically, 17 patients were lost after the initial blood sample, 21 after the first visit, 16 after the second visit, 8 after the third visit, and 4 after a total of five blood draws (Table 2).

### 3.2. Anti-HEV-IgM Seroprevalence

The seroprevalence measured using the Wantai assay was 0% in both the oncological and healthy cohorts (0/66, *p* = 1). Anti-HEV IgM seroprevalence, as determined by the Euroimmun assay, was 1% (1/66) in oncological patients, with a subgroup distribution of 2.5% in those with hematological malignancies and 0% in patients with solid tumors. In the healthy control group, the anti-HEV IgM seroprevalence was 0% (0/65) (*p* = 0.319). The Elecsys^®^ assay measured an IgM seroprevalence of 6% (4/66), comprising 10% in hematological malignancies and 0% in solid tumors.

### 3.3. Anti-HEV-IgG Seroprevalence

The HEV IgG seroprevalence determined using the Wantai assay was 38% (25/66) in the immunocompromised hematooncological patient group, with 40% (16/40) among patients with hematological malignancies and 35% (9/26) in those with solid tumors. In comparison, the seroprevalence in the healthy control group was 22% (14/65) (*p* = 0.041) (Table 3).

Using the Euroimmun assay, an overall IgG seroprevalence of 12% (8/66) was observed, with subgroup-specific rates of 15% (6/40) in hematological malignancies and 8% (2/26) in patients with solid tumors. In the healthy control cohort, IgG seroprevalence was 8% (5/65, *p* = 0.397).

Using the Elecsys^®^ assay, the seroprevalence in oncological patients was determined to be 53% (35/66), with a seroprevalence of 55% (22/40) in patients with hematological malignancies and 50% in those with (13/26) in solid tumors.

### 3.4. Subgroup Lymphoma Patients

We divided the cohort of oncology patients into two subgroups: lymphoma and non- lymphoma patients. A total of 42% (28/66) were diagnosed with lymphoma, while 58% (38/66) were non-lymphoma patients. None of the patients exhibited significantly elevated liver enzyme levels.

### 3.5. Concordance Analysis

Among the 66 samples that tested negative with the Wantai HEV-IgM assay, one sample (1%) was positive with the Euroimmun assay. The agreement between these two assays was slight (κ = 0.000). Similarly, four samples (6%) that were negative with the Wantai HEV-IgM assay tested positive in the Elecsys® assay, with slight agreement observed between these assays as well (κ = 0.000). In contrast, a fair level of agreement (κ = 0.385, *p* < 0.001) was observed between the Elecsys® and Euroimmun assays (Table 4).

A comparison of the positive test results between the Wantai HEV-IgG ELISA and the Euroimmun assay revealed that 16/ 24 samples (66.7%) tested positive using the Wantai assay and were negative in the Euroimmun assay, indicating a fair level of agreement (κ = 0.344, *p* < 0.001) between the two assays. Among 40 samples that tested negative using the Wantai assay, 10/ 40 (25%) were found to be positive in the Elecsys^®^ assay.

Cohen’s Kappa analysis demonstrated moderate agreement between the Wantai HEV-IgG and Elecsys® IgG assays (κ = 0.553, *p* < 0.001), while the agreement between the Elecsys® and Euroimmun assays was slight (κ = 0.190, *p* = 0.008).

### 3.6. Questionnaire

The response rate for the questionnaire was 96.97% (64/66). Linear regression analysis was performed using the serological data measured with the established Wantai Assay (Figure 1). Individuals employed in agricultural work but not involving animal farming were 5 times more likely to test positive for HEV antibodies (OR 5.317, 95% CI 0.118–239.249). Patients who professionally dealt with wastewater showed a reduced risk of anti-HEV-IgG positivity (OR 0.326, 95% CI 0.003–34.523), while the consumption of cooked pork at least once a year increases the risk by 64.5% (OR 1.645, 95% CI 0.190–14.235). The consumption of raw or undercooked pork more than once per month was associated with a 4-fold increase in risk (OR 4.526, 95% CI, 0.389–52.695). A consumption of 6–12 times per year indicated the highest overall increased risk by a factor of 6 (OR 6.367, 95% 0.545–74.447). The ingestion of venison increased the risk of anti-HEV-IgG activity by 41.5% (OR 1.415, 95% CI 0.338–5.929). Both the consumption of rocket or spinach (OR 0.822, CI 95% 0.155–4.361) and the consumption of alcohol (OR 0.641, 95% CI 0.065–6.283) appear to contribute to a risk reduction. Gender was not found to be a statistically relevant factor for anti-HEV IgG positivity (OR 0.703, 95% CI 0.128–3.852). Increasing age showed a modest risk increase of approximately 2% for anti-HEV IgG positivity (OR 1.027, 95% CI 0.963–1.096). The presence of an underlying oncological (cancer-related) disease was associated with a 15% increased risk of being anti-HEV IgG positive (OR 1.154, 95% CI 0.287–4.642).

## 4. Discussion

Numerous studies have highlighted discrepancies in the prevalence of anti-HEV antibodies due to the choice of assay methods used [34,35]. These analyses were almost exclusively confined to immunocompetent patients and data in cancer patients is lacking. Evidence suggests that the actual prevalence of HEV exposure within different population groups may be significantly underestimated depending on the serological methods used [29,36]. Immunocompromised patients are particularly at increased risk of developing chronic hepatitis E infections [5,13]. However, studies focusing on oncological patients with immunosuppression are scarce, underscoring the importance of addressing this gap.

Our study aimed to evaluate the reliability of three different anti-HEV assay methods (Wantai, Euroimmun, Elecsys^®^) in a prospective study design involving a cohort of immunosuppressed oncology patients. For comparison, a retrospective analysis of a healthy control group was included. Additionally, a questionnaire was provided to the oncology cohort to explore known risk factors for HEV infection in immunocompromised individuals. These efforts were intended to provide a more comprehensive understanding of HEV seroprevalence and its associated factors in a vulnerable population.

Initially, our goal was to examine the sensitivity of two commonly used ELISA-based assays in an oncology immunosuppressed cohort and determine their concordance. The Wantai assay is regarded as the serological “gold standard” in hepatitis E diagnostics due to its high detection rates, as evidenced by significantly higher seroprevalence rates across all cohorts in meta-analyses [37,38,39]. By contrast, the Euroimmun ELISA assay is an established and frequently utilized method in routine diagnostics in Germany [40].

In our study, the Wantai HEV-IgM assay showed no positives, while the Euroimmun assay detected one positive case in the oncology group, 1% (1/66). Concordance was only slight (κ = 0.000). For IgG, seroprevalences in the oncology cohort were higher using the Wantai assay: 38% (25/66), in comparison to 22% (14/65) (*p* = 0.041) in the healthy controls, with the highest detection rates in patients with hematologic malignancies, 40% (16/40). These findings suggest a higher prevalence in immunocompromised patients. In comparison, the Euroimmun assay showed lower seroprevalences in both groups (oncology: 12%, 8/66; controls: 8%, 5/65; *p* = 0.319), with the highest in those with hematologic malignancies (15%, (6/40)). These results indicate lower prevalences in the Euroimmun assay and poor concordance, between the applied assays, which is also illustrated by the only fair agreement (κ = 0.344, *p* < 0.001) using the Cohen’s Kappa coefficient and which aligns with previous studies. Al-Absi et al. reported that the Wantai IgG ELISA demonstrated the highest detection rate (18%) among five commercial HEV assays, compared to only 10.1% for the Euroimmun assay [41]. One of the hypotheses explaining these discrepancies includes the idea that the high titers reported by the Wantai assay are biologically implausible, possibly reflecting low test specificity [39]. Like other commercial assays, the Wantai assay has not been fully evaluated for specificity concerning past infections [39]. Applying these concerns to our findings, it remains unclear whether the higher test positivity in the Wantai Assay is due to increased sensitivity, particularly in immunocompromised patients, or whether it reflects lower specificity.

Given these challenges, we incorporated a further, third diagnostic test, the Elecsys^®^ immunoassay, to assess its performance in the at-risk immunosuppressed patient group. To our knowledge, no published studies have directly compared the Elecsys^®^ immunoassay with other HEV immunoassays. Therefore, we provide an initial evaluation of Elecsys^®^ in comparison with established HEV assays.

The Elecsys^®^ detected an overall anti-HEV-IgM prevalence of 6% (4/66) in the oncology cohort. Due to the frequent use of B-cell-depleting therapies, it is well established that lymphoma patients have an increased risk of infections. Therefore, we specifically analyzed the subgroup of patients diagnosed with lymphoma [42,43]. Among lymphoma patients (n = 28), 14% (4/28) tested positive for anti-HEV-IgM, while no positives were found in non-lymphoma patients (0%, 0/38, *p* = 0.028). Notably, these IgM-positive lymphoma cases did not exhibit significant liver enzyme elevations, making acute hepatitis E infection unlikely. Instead, false-positive serology in lymphoma patients has been addressed in several studies. Immunosuppression in lymphoma patients can complicate HEV serology and lead to false positive results. This phenomenon is partly attributed to altered immune responses in these patients and potential cross-reactivity of the assays, which has been observed with Epstein-Barr virus (EBV), Lyme disease, and HIV [44,45,46,47]. Additionally, cross-reactivities in anti-HEV-IgM assays with cytomegalovirus and Epstein-Barr virus antibodies have also been reported [48].

The Elecsys^®^ assay measured elevated seroprevalence rates, which we believe should also be critically evaluated. The overall IgG positivity in the cancer patient cohort was found to be 53% (35/66), including up to 55% (22/40) in patients with hematologic malignancies and 50% (13/26) in those with solid tumors. A study published earlier this year reported that automated chemiluminescence immunoassays can be affected by acute infections such as EBV and CMV, potentially influencing HEV serodiagnostics [49]. Similarly, false positive serological results have been documented in HIV patients using the Elecsys^®^ HIV assay [50].

Our findings demonstrate that, irrespective of the assay utilized, IgM and IgG seroprevalence is elevated in oncological patients. Existing studies consistently indicate a higher HEV seroprevalence among cancer patients compared to healthy controls. Bai et al. reported a significantly increased HEV seroprevalence in cancer patients (26.0%) compared to controls (13.0%) in eastern China, with leukemia patients exhibiting the highest rates [51]. Similarly, Lin et al. observed greater HEV seropositivity in cancer patients (46.36%) than in non-cancer individuals (32.50%) [52]. Furthermore, Chiu et al. identified an HEV IgG positivity rate of 16% among cancer patients in the United States, including two cases of chronic HEV infection in individuals with hematological malignancies [43]. Collectively, these findings suggest that cancer patients, particularly those with hematological malignancies, may have an increased susceptibility to HEV infection and its associated complications. This finding underscores the necessity of directly utilizing molecular biological testing methods (PCR) in oncology immunocompromised patients when there is clinical suspicion of an HEV infection, such as elevated liver enzyme levels [41] [53]. In particular, false-negative HEV serological results can significantly impact clinical decision-making and outcomes in oncology patients. A missed diagnosis may lead to unrecognized viral persistence, potentially resulting in progressive liver damage or liver failure [13,54,55].

To further analyze HEV risk factors and to better interpret the observed elevated seroprevalence rates measured by the Wantai assay in our cohort, we employed a questionnaire (Figure 1). Here, a significant risk increase was associated with the regular consumption of raw pork. Consuming raw pork 6–12 times per year showed the highest overall risk increase, with an odds ratio of 6 (OR 6.367, 95% CI: 0.545–74.447). Additionally, individuals working in agriculture without direct livestock contact exhibited a fivefold increased risk of testing positive for hepatitis E antibodies (OR 5.317, 95% CI: 0.118–239.249). This finding is in line with established risk factors.

Given that the highest IgM prevalence was observed using the Elecsys^®^ assay, we reviewed the individual questionnaires of participants who tested IgM positive. Half reported regular raw pork consumption, and all four indicated regular consumption of cooked pork. One individual reported agricultural employment without livestock exposure. These observations align with the existing literature. Zoonotic transmission has been identified as the primary source of infection for HEV genotype 3, which is particularly prevalent in Western industrialized countries [56,57]. Pork consumption, especially raw pork, has been described as a significant risk factor for hepatitis E infection [5]. The marked association between agricultural work and HEV seropositivity is a noteworthy finding of this study, particularly in the state of Brandenburg, where approximately 45% of the land is dedicated to agriculture, including diverse crop cultivation such as fruits and vegetables [58,59]. Studies have demonstrated that individuals with occupational exposure—such as veterinarians, agricultural workers, and hunters—have a higher susceptibility to HEV infection compared to the general population [60]. The use of fertilizers derived from animal feces has been identified as a potential source of infection, enabling indirect transmission through contaminated fruits and vegetables [7,8,9].

Our study has certain limitations that need to be addressed: The samples were obtained from a single medical center, resulting in a relatively small sample size. Another key limitation is the difference in age and gender distribution between the immunocompromised and control groups, as age is known to impact seropositivity. However, in our study, we observed only a slight increase in seropositivity with increasing age. Additionally, several oncological patients were lost to follow-up due to disease progression or adverse events, which is, however, a common occurrence in longitudinal studies involving cancer patients with high morbidity and mortality.

Despite these limitations, the strengths of our study lie in its prospective design, the inclusion of an immunosuppressed oncology cohort, and the utilization of three different diagnostic methods alongside a tailored questionnaire. These elements provide valuable insights into the complex interplay between diagnostic methods, patient characteristics, and HEV-associated risk factors in a high-risk population.

## 5. Conclusion

In conclusion, the exact prevalence of HEV antibodies remains uncertain. Serological assays showed unreliable results and oncology patients seem to exhibit higher seroprevalence rates compared to healthy controls. The limited concordance between the HEV assays used and the possible false-positive seroprevalence, particularly among lymphoma patients, underscores the necessity of molecular testing using PCR for oncological and, especially, immunosuppressed patients. By integrating serological, molecular, and risk factor analyses, future research can better address the challenges of accurate HEV diagnosis and prevention in vulnerable populations.

## Figures and Tables

**Figure 1 antibodies-14-00027-f001:**
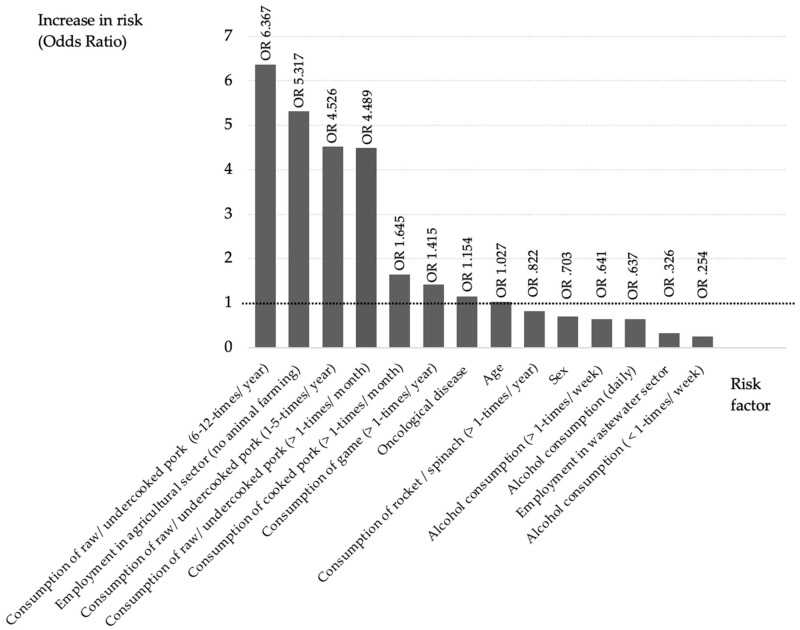
Odds ratio distribution depending on the risk factors. The black line illustrates an increase in risk. Abbreviations: OR = Odds ratio; n.a. = not applicable.

**Table 1 antibodies-14-00027-t001:** Demographic and clinical data (absolute and relative numbers) of both study populations including oncological patients and healthy controls.

	Oncological Disease (n = 66)	Healthy Controls (n = 65)	*p*-Value
Median age in years (SD)	72 (±11.4)	41 (±14.8)	<0.0001
Sex: n (%)			
Male	45 (68)	28 (43)	<0.0001
Female	21 (32)	37 (57)	<0.0001
Type of oncological disease: n (%)			
Hematological malignancy	40 (61)	n.a.	
Solid tumors	26 (39)		

Abbreviations: SD = standard deviation; n.a. = not applicable.

**Table 2 antibodies-14-00027-t002:** Overview of the collected blood samples in the oncological population.

	Hematological Malignant (n = 40)	Solid Tumors (n = 26)	Total (n= 66)
Initial blood collection	40	26	66
Lost to follow up			
after first collection	8	9	17
After 1st follow up	17	4	21
After 2nd follow up	9	7	16
After 3rd follow up	3	5	8
After 4th follow up	3	1	4

**Table 3 antibodies-14-00027-t003:** Seroprevalence for HEV in absolute and relative numbers using Wantai, Euroimmun and Elecsys^®^ Assay in all study populations.

	Oncological Disease (n = 66)	Healthy Controls (n = 65)	*p* Value
Wantai Assay: n (%)			
Anti-HEV-IgM	0/66 (0%)	0/65 (0%)	*p* = 1
Anti-HEV-IgG	25/66 (38%)	14/65 (22%)	*p* = 0.041
Hematological malignancy	16/40 (40%)		
Solid tumors	9/26 (35%)		
Euroimmun Assay: n (%)			
Anti-HEV-IgM	1/66 (1%)	0/66 (0%)	*p* = 0.319
Hematological malignancy	1/40 (2.5%)		
Solid tumors	0/26 (0%)		
Anti-HEV-IgG	8/66 (12%)	5/65 (8%)	*p* = 0.397
Hematological malignancy	6/40 (15%)		
Solid tumors	2/26 (8%)		
Elecsys ^®^ Assay: n (%)			
Anti-HEV-IgM	4/66 (6%)		
Hematological malignancy	4/40 (10%)		
Solid tumors	0/26 (0%)		
Anti-HEV-IgG	35/66 (53%)		
Hematological malignancy	22/40 (55%)		
Solid tumors	13/26 (50%)		

**Table 4 antibodies-14-00027-t004:** Agreement of qualitative results using Cohen’s Kappa Coefficient.

	Cohen’s Kappa Coefficient	*p* Value	Strenght of Agreement
Anti-HEV-IgM			
Elecsys^®^—Euroimmun	0.385	<0.001	fair
Elecsys^®^—Wantai	0.000		slight
Wantai—Euroimmun	0.000		slight
Anti-HEV-IgG			
Elecsys^®^—Euroimmun	0.190	=0.008	slight
Elecsys^®^—Wantai	0.553	<0.001	moderate
Wantai—Euroimmun	0.344	<0.001	fair

Kappa values interpreted as proposed by Landis and Koch, 1977 [33].

## Data Availability

Data presented in this study was extracted by I.-E.H.; the datasets of this study may be available from the corresponding author upon reasonable request.

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
