# Peer review of "Low Serological Agreement of Hepatitis E in Immunocompromised Cancer Patients: A Comparative Study of Three Anti-HEV Assays"

_2073-4468, 2025, doi:10.3390/antib14020027_

Round 1

Reviewer 1 Report

Comments and Suggestions for Authors

The article written by Haller I.E. and entitled "Low Serological Agreement of Hepatitis E in Immunocompromised Cancer Patients: A Comparative Study of Three Anti-HEV Assays" is well presented and easy to read and understand. The main objective is clear and it  aimed to compare the reliability and concordance of three HEV antibody assays Wantai, Euroimmun, and Elecsys in immunosuppressed oncology patients. Authors concluded that Oncology patients seem to exhibit higher seroprevalence rates compared to healthy controls and that The limited concordance between the HEV assays used and the possible false-positive seroprevalence, particularly among lymphoma patients, underscore the necessity of molecular testing using PCR for oncological and, especially, immunosuppressed patients. 

The methodology used and selected population are in coherence with obtained results and formulated discussion. This work is with importance for laboratory assays especially sereological tests used for the detection and the control of infected HEV patients.

Reviewer 2 Report

Comments and Suggestions for Authors

The publication “Low serological agreement of hepatitis E in immunocompromised cancer patients: a comparative study of three anti-HEV assays” is devoted to the comparison of the results of three serologic tests of different manufacturers on a cohort of cancer and healthy patients. Among the main drawbacks the following can be identified:

  1. The comparison includes two tests performed by one technology (Wantai and Euroimmun) and one test performed by another (Elecsys). It is incorrect to compare tests with different approaches to each other in case of this article.
  2. It is also unclear why the Wantai test is chosen as the “gold standard”. The effectiveness of this test is not confirmed in this article by any other tests (PCR, etc.). As a result, after reading the article, it remains unclear which test gave the most accurate results.
  3. The sample of patients is too small and too narrow (especially the selection of healthy patients only among health workers). In addition, the sample of cancer patients is unclear: patients with what diagnoses are present there?
  4. Obviously, on patients with oncohematologic diseases, which often result in abnormal serologic assays, the tests will show unreliable results. The authors obtain just such results.
  5. Such works have already been published quite a lot, as the authors themselves mention. Thus, the scientific value of this publication is rather low.
  6. Confirmation of IgG titer in patients with cancer does not indicate that they have these antibodies because they have been immunologically compromised; they may have had hepatitis E virus infection earlier, before cancer manifestation. Considering the rather aged sample of patients with cancer, this is quite possible. It is unclear whether the authors took this into account.
  7. The lack of increased risk of hepatitis E virus infection in association with cancer identified by the authors strangely does not correlate with the introduction to this article describing the risks for such patients.
  8. In Section 2.2. it is not clear why positive and negative values are defined differently for different tests, at different cutoffs.
  9. In addition, the text of the article is rife with typos (e.g., line 261 - Germany with a lowercase letter; 269 - prevalence and immunocompromised). Table 4 and 5 are arranged in a strange, incomprehensible way for the reader.

Thus, the article gives a negative impression: it is not clear what conclusion the reader should draw from it. The patient sample is poor, there is no comparison of serologic test results with the more accurate laboratory evaluation of active disease or antibodies, and it is not clear which test is credible in the end. In addition, the sampling of cancer patients and the point of studying tests with unclear reliability on cancer patients is unclear. 

Reviewer 3 Report

Comments and Suggestions for Authors

General comments: In this study, Haller et al. examined the reliability and concordance of three HEV serological assays, including Wantai, Euroimmun, and Elecsys, in immunosuppressed oncology patients. The study analyzed data from 66 oncology patients between September 2020 and 2021 and 65 healthy controls. HEV IgM and IgG seroprevalence varied across assays, with patients with hematological malignancies showing highest IgG seroprevalence. Additionally, using questionnaires and risk factor analysis, the authors found that the highest association between HEV exposure and the consumption of undercooked pork or crop-based agriculture. While the topic is relevant, certain aspects of the analysis need further clarification, and the presentation of results could be improved. I have the following comments that the authors may consider addressing:

Specific comments:

  1. Abstract, line 29: Did the author quantify anti-HEV IgM and IgG antibodies in all samples? It may be more accurate to use the term “tested” instead.
  2. Section 2.1: As far as I understand, none of these oncology patients had chronic hepatitis E. Could the author please confirm this?
  3. Section 3.3 and Table 3: What were the results for HEV IgG seroprevalence in healthy controls? If they were not tested, could the authors clarify why not?
  4. Notably, Table 3 and Table 5 are not referred to in the main text.
  5. Table 4 and Table 5 are difficult to read. I suggest omitting these two Tables since the results are already presented in Table 3.
  6. Section 3.4: Are there any correlations between the OD values and COI in the Wantai, Euroimmun, and Elecys assays? For example, do those 16/ 24 samples (66.7%) tested positive by the Wantai assay but negative with the Euroimmun assay exhibit lower OD values?
  7. Figure 1 is not referred in the main text.
  8. Section 3.5 confused this reviewer. Are the data presented from individuals with oncological diseases or healthy controls? Additionally, please specify the number of datasets used in the analysis. Finally, the English version of the questionnaire should be included in the Supplementary Materials.
  9. Discussion section, lines 262 to 276: This part is repetitive with the Results section. I recommend shortening this part to avoid redundancy.
  10. Discussion section, line 290: The sentence should be softened, as it is unclear whether a comparison between Elecsys and other HEV detection assays has been conducted but remains unpublished.
  11. Discussion section, line 295: This statement is not described in the Results section.
  12. Discussion section: The authors should also note the limitations of samples in a single medical center with a relatively small sample size.
  13. Discussion section: It would be beneficial for the authors to compare the HEV seroprevalence observed in oncology patients and healthy controls in this study with that from other similar studies.
  14. There are several typos and grammatic errors throughout the manuscript. English proofreading is recommended.

Minor comments:

  1. Line 158: Delete “ongoing”.
  2. Line 218: “with”.
  3. Line 220: change “hepatitis E virus” to “HEV”.
  4. Line 380: “hepatitis E virus”.
  5. Line 242: “and”.
  6. Line 441: Please check the format of the reference 42.

Reviewer 4 Report

Comments and Suggestions for Authors

This study investigates the reliability and concordance of three anti-HEV serological assays—Wantai, Euroimmun, and Elecsys—in immunocompromised oncology patients. The authors highlight the significant variability among these assays, emphasizing the difficulty of obtaining consistent HEV seroprevalence estimates in this vulnerable population. Their findings underscore the need for careful assay selection and improved diagnostic strategies to ensure accurate HEV screening in immunosuppressed individuals.

The paper presents an important contribution to the field of infectious diseases and oncology, particularly given the increasing recognition of chronic HEV infections in immunocompromised patients. However, the discussion could benefit from a more detailed exploration of the clinical implications of discordant assay results. For instance, how might false-negative or false-positive findings influence clinical decision-making in this patient group? Additionally, further consideration of whether molecular testing (e.g., HEV RNA detection) could serve as a complementary diagnostic tool would strengthen the study's conclusions.

From a language perspective, the manuscript is generally well-written but could be improved by enhancing clarity and conciseness in certain sections. For example, the phrase "variability in serological assay performance complicates the assessment of HEV antibody prevalence in these at-risk populations" could be simplified to "assay variability complicates HEV prevalence assessment in at-risk groups." Small refinements like this would improve readability. Additionally, minor grammatical inconsistencies, such as missing articles and slight awkwardness in phrasing (e.g., "Samples were collected both at baseline (treatment-naive) and during ongoing treatment (on therapy)"), could be adjusted for smoother flow.

Overall, the study is well-conceived and methodologically sound, but a more in-depth discussion of its clinical relevance and diagnostic implications would further enhance its impact.

Comments on the Quality of English Language

From a language perspective, the manuscript is generally well-written but could be improved by enhancing clarity and conciseness in certain sections. For example, the phrase "variability in serological assay performance complicates the assessment of HEV antibody prevalence in these at-risk populations" could be simplified to "assay variability complicates HEV prevalence assessment in at-risk groups." Small refinements like this would improve readability. Additionally, minor grammatical inconsistencies, such as missing articles and slight awkwardness in phrasing (e.g., "Samples were collected both at baseline (treatment-naive) and during ongoing treatment (on therapy)"), could be adjusted for smoother flow.

Round 2

Reviewer 2 Report

Comments and Suggestions for Authors

The authors have extensively revised the article. All comments have been taken into account and the text of the publication has been supplemented with the necessary information. It should also be noted that the authors have written an impressive point-by-point response to the comments. This text contains a large number of references to publications that would have been appropriate in the text of the main article and would have supported many of the authors' claims. I recommend that the authors add these references to the article. In this way, the article can be returned to the authors for minor revision.

Author Response

The authors have extensively revised the article. All comments have been taken into account and the text of the publication has been supplemented with the necessary information. It should also be noted that the authors have written an impressive point-by-point response to the comments. This text contains a large number of references to publications that would have been appropriate in the text of the main article and would have supported many of the authors' claims. I recommend that the authors add these references to the article. In this way, the article can be returned to the authors for minor revision.

Dear Reviewer,

We are delighted to receive your positive feedback. Once again, we would like to thank you for your effort and valuable comments. In accordance with your suggestions, we have expanded our manuscript by incorporating numerous references cited in our response.

We are pleased to present you with our revised manuscript.

Yours sincerely,

Isabel-Elena Haller (for all authors)

Reviewer 3 Report

Comments and Suggestions for Authors

I am satisfied with the revised manuscript, as the authors have addressed most of my concerns. However, regarding comment 8, I am still unable to locate the questionnaire. I believe it is crucial for readers to have access to it in order to better understand the study. Could you kindly confirm its inclusion? Additionally, it is uncommon for Figure 1 to first appear in the Discussion section. It would be more appropriate to describe it in the Results section.

Author Response

I am satisfied with the revised manuscript, as the authors have addressed most of my concerns. However, regarding comment 8, I am still unable to locate the questionnaire. I believe it is crucial for readers to have access to it in order to better understand the study. Could you kindly confirm its inclusion? Additionally, it is uncommon for Figure 1 to first appear in the Discussion section. It would be more appropriate to describe it in the Results section.

Dear Reviewer,

We are delighted to receive your positive feedback. Once again, we would like to thank you for your effort and valuable comments.

The questionnaire was included in the submitted ZIP file along with the manuscript. Additionally, we have now uploaded it again as part of the supplementary files. I hope you are now able to access it.

We have provided an explanation for Figure 1 in Section 3.6. To further clarify this, we have added a corresponding note in the text (line 225).

We are pleased to present you with our revised manuscript.

Yours sincerely,

Isabel-Elena Haller (for all authors)